# Reducing Seed Shattering in Weedy Rice by Editing *SH4* and *qSH1* Genes: Implications in Environmental Biosafety and Weed Control through Transgene Mitigation

**DOI:** 10.3390/biology11121823

**Published:** 2022-12-14

**Authors:** Yu-Liang Zhang, Qi-Yu Xia, Xiao-Qi Jiang, Wei Hu, Xiao-Xue Ye, Qi-Xing Huang, Si-Bin Yu, An-Ping Guo, Bao-Rong Lu

**Affiliations:** 1National Key Laboratory of Crop Genetic Improvement, National of Plant Gene Research (Wuhan), Huazhong Agricultural University, Wuhan 430070, China; 2Hainan Key Laboratory for Biosafety Monitoring and Molecular Breeding in Off-Season Reproduction Regions, Institute of Tropical Bioscience and Biotechnology, Chinese Academy of Tropical Agricultural Sciences, Haikou 571101, China; 3Hainan Yazhou Bay Seed Laboratory, Sanya Research Institute of Chinese Academy of Tropical Agricultural Sciences, Sanya 572000, China; 4Ministry of Education Key Laboratory for Biodiversity and Ecological Engineering, Fudan University, Shanghai 200433, China

**Keywords:** *Oryza sativa* f. *spontanea*, gene expression, seed persistence, transcriptome, transgene mitigation, weed

## Abstract

**Simple Summary:**

Mitigating the possible adverse environmental impacts caused by transgene flow from genetically engineered crops to their wild/weedy relatives is an ideal strategy for resolving biosafety problems. To explore a transgene mitigation system in rice, we edited the seed-shattering genes (*SH4* and *qSH1*) of a weedy rice line with strong seed shattering. The results showed substantially reduced seed shattering in the gene-edited weedy rice lines. The single gene-edited weedy rice lines displayed an inconsistent reduction in seed size-related traits. Reduced seed shattering was closely linked with a lack of abscission layers and reduced abscisic acid (ABA). In addition, most examined genes that were closely associated with ABA biosynthesis, ABA signaling transduction, and cell wall hydrolysis were downregulated in the gene-edited weedy rice lines. These findings can assist in exploring the underlying mechanisms of reduced seed shattering in weedy rice plants, in addition to practical applications for mitigating environmental impacts caused by transgene flow and for controlling the infestation of weedy rice.

**Abstract:**

Mitigating the function of acquired transgenes in crop wild/weedy relatives can provide an ideal strategy to reduce the possible undesired environmental impacts of pollen-mediated transgene flow from genetically engineered (GE) crops. To explore a transgene mitigation system in rice, we edited the seed-shattering genes, *SH4* and *qSH1*, using a weedy rice line (“C9”) that originally had strong seed shattering. We also analyzed seed size-related traits, the total genomic transcriptomic data, and RT-qPCR expression of the *SH4* or *qSH1* gene-edited and *SH4*/*qSH1* gene-edited weedy rice lines. Substantially reduced seed shattering was observed in all gene-edited weedy rice lines. The single gene-edited weedy rice lines, either the *SH4* or *qSH1* gene, did not show a consistent reduction in their seed size-related traits. In addition, reduced seed shattering was closely linked with the weakness and absence of abscission layers and reduced abscisic acid (ABA). Additionally, the genes closely associated with ABA biosynthesis and signaling transduction, as well as cell-wall hydrolysis, were downregulated in all gene-edited weedy rice lines. These findings facilitate our deep insights into the underlying mechanisms of reduced seed shattering in plants in the rice genus *Oryza*. In addition, such a mitigating technology also has practical applications for reducing the potential adverse environmental impacts caused by transgene flow and for managing the infestation of weedy rice by acquiring the mitigator from GE rice cultivars through natural gene flow.

## 1. Introduction

With the rapid development of biotechnology, genetically engineered (GE) plants have been extensively cultivated around the globe [1], which has aroused worldwide concerns and even debates over biosafety issues, including undesired environmental impacts [2,3,4,5]. Of these, transgene escape from a GE plant to its wild/weedy relatives through pollen-mediated gene flow (PMGF) and its environmental impacts are the most debated issues. Under such scenarios, a transgene conferring GE novel traits with strong selective advantages, such as resistance to insects, diseases, weeds, and abiotic stresses, may significantly enhance the fitness of PMGF hybrids and their progeny, possibly causing undesired environmental impacts [6,7,8]. In fact, many studies have confirmed the likelihood of transgene flow from a crop species to its wild relatives, particularly to conspecific weeds that belong to the same biological species of the crop [8,9,10,11,12]. Studies have also indicated that it is nearly impossible to prevent transgene flow from a GE crop to its conspecific weeds, simply because the conspecific weeds do not have obvious reproductive barriers with their crop species [9], creating difficulties for farmers in appropriately managing such impacts caused by crop-to-weed transgene flow.

Weedy rice (*Oryza sativa* L. f. *spontanea* Rosch.), also referred to as red rice, is a noxious weed that infests cultivated rice fields worldwide and causes serious yield losses of cultivated rice (*Oryza sativa* L.) [13,14]. Weedy rice is a typical conspecific weed that belongs to the same species of cultivated rice. It is difficult to identify weedy rice from its crop, but it is characterized by strong seed shattering at maturity [14]. Many studies have reported PMGF from cultivated rice to weedy rice at various frequencies based on designed experiments or the analyses of extensive rice field sampling [15,16,17]. Controlled field experiments and large-scale rice field sampling have shown that the PMGF frequency from cultivated rice to weedy rice populations is largely variable but up to ~0.5% per generation in different locations [15,16,18]. Altogether, these results suggest that transgene movement from cultivated rice to weedy rice is not preventable in the field.

In fact, genes conveying traits with strong natural selective advantages, such as herbicide tolerance and moving from rice varieties to weedy/wild rice populations through PMGF, are frequently reported [5,7,17,19,20]. Such crop-to-weed/wild gene flow has caused tremendous problems for rice production. For example, the commercial release and production of herbicide-resistant rice varieties (Clearfield^®^, containing an artificial mutagenic imidazolinone-herbicide resistance gene) have aroused social economic problems for local famers in various locations, such as in the USA, Brazil, Greece, and Italy, due to the resulting herbicide-resistant weedy rice from crop-to-weed gene flow [21,22,23,24]. The rapid spread of a gene with strong natural selective advantages demonstrates the inevitable crop-to-weed transgene flow, which may cause serious problems for rice production and environmental biosafety. Therefore, it is extremely important and urgent to develop strategies to significantly reduce such adverse impacts, given that rice crop-to-weed transgene flow is not preventable [25].

Transgene mitigation (TM) is probably the only strategy for reducing the undesired environmental impacts of crop-to-weed transgene flow in rice [25]. TM is a recently proposed concept or strategy for reducing the unwanted environmental impacts caused by transgene flow from GE crops to their wild or weedy relatives [26]. The key point of the TM design and technique is to insert a tandem secondary mitigation gene or “mitigator” together with the target transgenes of interest into a GE crop [25,26,27]. The tandem mitigator can provide phenotypic traits with reduced fitness for wild or weedy relatives of crop plants (e.g., reduced seed shattering) but has little or no adverse effect on the crop plants (e.g., seed persistence), which can restrict or slow down the spread of the transgene in agroecosystems and reduce the unwanted environmental impacts. Although pollen-mediated weed-to-crop transgene flow is hardly preventable, the mitigator, serving as a “safe-box,” can counterbalance the fitness of the PMGF-resulting weedy plants and mitigate the unwanted environmental impacts.

In the case of weedy rice, reduced seed shattering is a domesticated characteristic that is fatal for the weeds, depending on natural seed shattering for self-reproduction, but beneficial for cultivated rice, for which seed persistence is critical for the grain harvest [14,28]. Therefore, reduced seed shattering has great potential for developing a transgene mitigator, a “safe-box,” to reduce the fitness benefit to weedy plants that acquire the transgene(s) through PMGF from GE rice varieties, safeguarding the application of GE rice. In addition, given that the ideal mitigator reduces the seed shattering offsets of any potential benefit conferred by crop genes possibly through PMGF and reduces the fitness of weed rice, the construction of GE rice with such a mitigator can also provide an effective strategy for weedy rice control through crop-to-weed PMGF in rice.

A previous study successfully developed two GE rice lines containing artificial micro-RNA genes by silencing the rice *SH4* gene, a QTL (quantitative trait locus) responsible for seed shattering in cultivated rice [25]. That study demonstrated significantly reduced seed-shattering ability in the hybrid plants and their F_2_ progeny (between the GE cultivated rice lines and weedy rice) containing the mitigators. In addition, a significantly reduced number of seeds from the hybrids and F_2_ progeny were disposed of in the soil seed banks [25]. However, whether it is possible to edit the seed-shattering genes, including *SH4*, in weedy rice to develop substantially reduced seed-shattering lines remains unknown. In addition, no investigation has attempted to explore the possibilities of editing more than one seed-shattering gene in weedy rice at the same time to meet the goal of transgene mitigation. Therefore, there is great potential to develop a “safe-box” system through gene editing in rice to ensure the biosafety of transgenic biotechnology.

In this study, we produced weedy rice plants by editing the *SH4* and *qSH1* genes individually and the combination *SH4*/*qSH1* genes using *CRISPR*-*Cas9* technology. We examined seed shattering and seed size-related traits of the parental weedy rice line and the gene-edited plants, in addition to their transcriptomic and metabolomic characteristics. The primary objectives of this study were to address the following questions: (i) Can editing of the *SH4* or *qSH1* gene or both of the genes substantially reduce seed shattering of weedy rice plants? (ii) Does editing of the seed shattering genes, *SH4* and *qSH1*, influence the seed-size related traits of the edited plants? (iii) What genes in the transcriptomic and metabolomic pathways are responsible for changes in the gene-edited weedy rice plants? The answers to these questions will facilitate our understanding of the underlying mechanisms for seed shattering and for designing effective transgene mitigator strategies to reduce the potential environmental impacts caused by transgene spread. In addition, the seed-shattering reduction system can also be useful for developing strategies to control weedy rice by producing GE rice varieties containing a mitigator that can naturally transfer to weedy rice and substantially reduce seed shattering.

## 2. Materials and Methods

### 2.1. Production of SH4 and qSH1 Gene-Edited Plants

A seed-shattering parental weedy rice line (“C9”) used for *SH4* and *qSH1* gene editing was collected from the Chidou Village in Guangdong Province, at the location of 20°52.500′ N, and 109°46.501′ E in China. In addition, a widely cultivated *indica* rice variety (‘Shouxiang-1′) in southern China was collected from the Yazhou District in Sanya City, Hainan Province and used as the control for seed persistence.

The first exons of the *SH4* and *qSH1* genes of “C9” were amplified for DNA sequencing. CRISPR primers were designed based on the obtained DNA sequences. Then, two single locus-targeting editing vectors, coded as pRLG103-SH4 (targeting on two loci, namely SH4-TL-a and SH4-TL-b located on *SH4*) and coded as pRLG103-qSH1 (targeting on two loci, namely qSH1-TL-a and qSH1-TL-b located on *qSH1*), were constructed. In addition, a double locus-targeting editing vector, coded as pRLG103-SH4/qSH1 (targeting on two loci, namely SH4-TL-a and qSH1-TL-b, located on *SH4* and *qSH1*, respectively), was also constructed. The vectors were sequenced and verified by Invitrogen Biotech Company (Shanghai, China).

After *Agrobacterium*-mediated transformation of the editing vectors, the DNA samples of the transformed seedlings were extracted with a Plant DNA Extraction Kit (Tiangen, China) and used for transgene amplification and editing target sequencing verification. The base nucleotide mutations were analyzed. Based on the results of the sequence, the selfed offspring T1 of the edited T0 plants with homozygous biallelic mutations were selected, of which the editing target locus was sequenced. The T1 plants with stable mutations were selected as the experimental lines (Table 1).

### 2.2. Measurement of Seed Shattering- and Seed Size-Related Traits of Gene-Edited and Control Lines

To assess the effect of *SH4* and *qSH1* gene editing on seed shattering of weedy rice, we examined the seed-shattering index (SSI) of ~100 seeds from 10 rice plants that were derived from each of the five *SH4*-gene edited lines, five *qSH1*-gene edited lines, three *SH4*/*qSH1*-double-gene edited lines, the parental weedy rice line (“C9”) as the seed-shattering control, and the cultivated *indica* rice variety (“Shouxiang-1”) as the seed persistence control. In the examination, pulling force was measured on the seed until it was shattered from the panicle, and the peak value of the pulling force, namely the breaking tensile strength (BTS), was measured using a tensile tester, following the method of Yan et al. (2017). The SSI was calculated following the formula SSI = e^−BTS^ × 100%. The SSI of the weedy rice control (“C9”) at maturity was set as 100%.

In addition, a few seed-size related traits, including 1000-seed weight, seed length, and seed width of weedy rice lines (parent and edited), were measured to assess the potential impact of gene editing on seed size associated with grain yield. To measure the 1000-seed weight, five replicates were included, each with 1000 seeds from mixed seeds of 10 plants using an automatic seed counter. To measure seed length and width, five replicates were included, each containing 20 seeds randomly sampled from the same mixed seed pool using a vernier caliper.

### 2.3. Electronic Microscopy of Pedicels of Gene-Edited and Parental Weedy Rice Lines

For scanning confocal microscopy examination, the samples, including the abscission layers with pedicels and spikelets of weedy rice, were collected 1–2 days before or at flowering. These samples included the *SH4* (S5), *qSH1* (Q5), and *SH4*/*qSH1* double gene edited (SQ1) lines, in addition to the weedy rice parent (“C9”). The samples were longitudinally cut and stained with 0.1% acridine orange solution for 15 min. The prepared samples were fixed on glass slides and then observed by the FV1000 laser scanning confocal microscope (Olympus Corp., Tokyo, Japan) under laser sources of 488 and 559 nm.

For scanning electronic microscopy examination, the pedicel samples were collected after maturity of the weedy rice seeds. The pedicel samples, including the abscission layers, were cut from the spikelet, soaked in the fixative solution containing 2.5% (pH 7.2–7.4) glutaraldehyde for more than 4 h, and rinsed with 0.1 m phosphate buffer. The prepared samples were dehydrated with ethanol, soaked in tertiary butanol for more than 15 min, and dried with a carbon dioxide critical point dryer. After removing the spikelet, the pedicel samples were conducted with vacuum ion sputtering coating and observed (the fractures surface facing up) using the TM4000 Plus scanning electron microscope (Hitachi Ltd., Tokyo, Japan).

### 2.4. Transcriptomic Sequencing of the Gene-Edited and Parental Weedy Rice Lines

For transcriptomic sequencing, ~500 pedicel samples from the gene-edited lines (S5, Q5, and SQ1) and weedy rice prenatal line (“C9”) were collected at 1, 15, and 25 days after heading, representing the preliminary, middle, and late stages of seed development, respectively. A total of three biological replicates were performed. Total RNA was extracted from the pedicel tissues with an RNA extraction kit (DP432, TIANGEN, Beijing, China) according to the manufacturer’s instructions. After purification with the TruSeq Stranded mRNA LTSample Prep Kit (Illumina, San Diego, CA, USA), the RNA samples were fragmented and used as a template for cDNA synthesis. The obtained cDNA samples were A-tailed, ligated, and sequenced (Appendix A) using the Aglient 2100 bioanalyzer (Agilent Technologies, Inc., Santa Clara, CA, USA).

Adaptor sequences and low-quality reads were removed using Trimmomatic v0.36 [29]. High-quality 150 bp paired-end reads were mapped on the reference genome, cultivated rice R498 (*Oryza sativa* ssp. *indica*) [30] with HISAT2 v2.2.1 and StringTie v1.3.3 [31]. Gene expression levels were represented by the FPKM (fragments per Kb per million reads) values.

### 2.5. Real-Time Quantitative PCR of Genes Related to the Seed Shattering of Weedy Rice

The relative expression quantities of shattering-controlling QTLs (*CPL1*, *sh4*, *SHAT1*, *OsSH1*, *OsSH15*, *qSH1*, *SH5*, *SH8*), genes encoding the key enzymes of ABA (abscisic acid) *ZEP*, *NCED1.1*, *NCED1.2*, *SDR1*, *SDR*), ABA-signaling-pathway-related genes (*PYL3*, *PYL4*, *PP2C39*, *SAPK5*, *BZIP23*), genes encoding the cell-wall hydrolases (*PME34*, *PG*, *XTH28*, *MAN4*), and reference gene *Ubiquitin* were examined using real-time quantitative PCR. The plant sample materials were collected at 1, 15, and 25 days after heading and then used for RNA extraction following the same method as transcriptomic sequencing.

Real-time PCR was performed using LightCycler^®^ 480 II Real-time PCR Instrument (Roche, Basel, Switzerland) with 10 μL PCR reaction mixture that included 1 μL cDNA, 5 μL of 2× *PerfectStart*^TM^ Green qPCR SuperMix, 0.2 μL forward primer, 0.2 μL reverse primer, and 3.6 μL water. The primers were designed referencing the target-gene sequence obtained from the NCBI database (Appendix A). The PCR reaction was conducted using the LightCycler^®^ 480 II Real-time PCR Instrument (Roche) with the following conditions: 30 s at 94 °C, followed by 45 cycles of 94 °C for 5 s and 60 °C for 30 s. The relative expression quantities of mRNA were normalized to the reference gene *Ubiquitin* and calculated using the 2^−ΔΔCt^ method [32], in which the content was determined as the multiple quantity (M) relative to the expression of the *SH4* gene in the weedy rice control (“C9”).

### 2.6. Examination of the Shattering-Related Hormone Contents of Gene-Edited and Parental Weedy Rice Lines

The contents of the shattering-related hormones, including ABA, IAAs (3-Indoleacetic acid, auxin a and auxin b), and cytokinins (tZ, cZ, tZR, and cZR), were examined using high-performance liquid chromatography tandem mass spectrometry (HPLC-MS). For the examination, ~80 mg of the pedicel tissues of the gene-edited lines (S5, Q5, and SQ1) and the weedy rice prenatal line (“C9”) were collected at 1, 15, and 25 days after heading, representing the preliminary, middle, and late stages of seed development, respectively. The collected pedicel tissue samples were mixed with 20 μL of 0.06 mg/mL L-2-chlorophenylalanine methanol solution and 1 mL of 70% methanol water solution, precooled to −20 °C, and ground. The mixture was ultrasonically extracted for 30 min and kept overnight at −20 °C.

After centrifugation, 2 μL of the extracting supernatant was filtrated and injected into the ACQUITY UPLC I-Class plus liquid chromatograph (Waters Corp., Milford, MA, USA). In the liquid chromatography test, 0.1% formic acid water solution was used as eluent solvent A, and 0.1% formic acid acetonitrile solution was used as eluent solvent B for the mobile phase. The concentration of the eluent started from 95% solvent to 0% solvent A after 14 min, and increased to 95% after 15.1 min. The concentration was then kept isocratic at 95% solvent A between 15.1 and 16 min. The QE plus mass spectrometer (Thermo Fisher Scientific, Waltham, MA, USA) was used for mass spectrometric analysis with the following electrospray ionization (ESI) parameters: spray voltage, 3800 V for the cation and −3000 V for the anion; capillary temperature, 320 °C; auxiliary gas heater temperature, 350 °C; sheath gas flow rate, 40 Arb for the cation and 35 Arb for the anion; and aux gas flow rate, 10 Arb for the cation and 8 Arb for the anion. The hormone content was determined as the multiple quantity (M) relative to the *SH4* gene expression in the weedy rice control (“C9”), using the Progenesis QI V2.3 (Nonlinear, Dynamics, Newcastle, U.K.).

### 2.7. Data Analysis

One-way ANOVA (analysis of variance) was conducted to estimate the effect of *SH4*, *qSH1*, and double-gene (*SH4/qSH1*) editing on seed shattering- and seed size-related traits. Duncan’s multiple range comparison was used to compare differences in seed shattering- and seed size-related traits among different lines [33]. The independent-sample Student’s *t*-test [34] was used to compare differences in expression quantities of the genes examined by real-time quantitative PCR and the hormone content between the gene-edited lines and the weedy rice parent. All statistical analyses were performed using SPSS ver. 22.0 (IBM Inc., New York, NY, USA). The differentially expressed genes (DEGs) detected by transcriptomic sequencing were determined by DEseq2 based on a threshold of two-fold expression change and p-value < 0.05 [35]. Gene ontology (GO) enrichment analysis was performed using the topGO v2.8 R package.

## 3. Results

### 3.1. Variation in Seed Shattering- and Seed Size-Related Traits in SH4- and qSH1-Edited Weedy Rice Plants

One-way ANOVA showed that editing of a single gene, either *SH4* or *qSH1*, or two genes, *SH4*/*qSH1* (double editing), had significant effects on the SSI compared with the weedy rice control, “C9” (Table 2). These results suggest the possibility of reduced seed shattering in weedy rice lines by editing these seed-shattering genes. In addition, editing of these genes also had significant effects on the seed length of the edited weedy rice lines compared with “C9” (Table 2), although editing of the single *SH4* or *qSH1* gene did not have significant effects on seed width or 1000-seed weight (*p* ≥ 0.05). However, different gene-edited weedy rice lines (e.g., S1, S2, S3, S4, S5, Q1, Q2, Q3, Q4, Q5, SQ1, SQ2, and SQ3) had more complicated patterns for the examined traits, such as the SSI, 1000-seed weight, seed length, and seed width (*p* < 0.001, Table 2), suggesting significant variations in the examined traits among different gene-edited weedy rice lines.

Further analysis of seed shattering by Duncan’s multiple range test, involving the seed-shattering control weedy rice “C9”, a cultivated rice line (used as the seed persistence control), and the gene (*SH4*, *qSH1*, and *SH4*/*qSH1*)-edited lines derived from the “C9” control, showed significant decreases (*p* < 0.05) in the SSI of all gene-edited lines (Figure 1). However, no significant differences were detected among different lines with the same edited gene. These results demonstrated that editing of the seed shattering genes, either a single gene (*SH4* or *qSH1*) or two genes (*SH4*/*qSH1*), substantially reduced shattering of weedy rice seeds with a consistent effect among different gene-edited lines/plants (*p* < 0.05, Figure 1a–c). Noticeably, the reduced seed shattering ability (or seed persistence ability) in the gene-edited lines was even more pronounced (*p* < 0.05) than that of the seed persistent cultivated rice control (gray columns in Figure 1a–c).

In addition, results from the analyses of seed size-related traits by Duncan’s multiple range test did not show a significant reduction (*p* ≥ 0.05) in 1000-seed weight between the weedy rice control “C9” and the single *SH4* or *qSH1* gene-edited lines (Figure 2a,b). However, a significant reduction (*p* ≥ 0.05) in 1000-seed weight was detected in the double gene (*SH4*/*qSH1*)-edited lines compared with the weedy rice control “C9” (Figure 2c). These findings indicate that editing a single gene, either the *SH4* or *qSH1* gene, did not considerably affect the seed weight of weedy rice. Furthermore, the results of Duncan’s multiple range test further showed significantly reduced seed length (*p* < 0.05) in nearly all gene-edited weedy rice lines for both single (*SH4*, *qSH1*) and double gene-edited (*SH4*/*qSH1*) lines (Figure 3a–c). However, no significant differences (*p* ≥ 0.05) in seed width were observed in most of the gene-edited lines (Figure 4a–c). These findings indicate the considerable influences of gene editing on seed length but very little influence on 1000-seed weight and seed width.

To understand the phenotypic changes before and after gene editing, we conducted a laser scanning confocal microscope examination of rice pedicels at the longitudinal section of seed-shattering control and seed-shattering reduced plants. The results showed great differences in the formation of the abscission layers (Figure 5). The shattering control of weedy rice “C9”, which had the strongest seed shattering, showed an obvious abscission layer with completely hydrolyzed cell wall tissues (Figure 5d). Such a formation of the abscission layer indicated the ease with which the pedicels break, which could cause strong seed shattering of the weedy rice control “C9”. For some gene-edited plants, such as S5, in which the *SH4* gene was edited to lose the seed-shattering function, reduced abscission layers with somewhat hydrolyzed cell wall tissues were observed (Figure 5a) in the pedicels, corresponding to the phenotypes of some degrees of reduced seed shattering. However, for other gene-edited plants, of which the *qSH1* (e.g., Q5) gene or both the *SH4* and *qSH1* (e.g., SQ1) genes were edited, no abscission layers or hydrolyzed cell wall tissues were observed in the pedicels (Figure 5b,c), corresponding to greatly reduced seed shattering. The former case (S5) was very similar to those of the cultivated rice control, which showed weak seed shattering (Figure 5e).

Similarly, scanning electron microscope examination on the intersecting surface of the rice pedicels after the seeds were shattered also provided evidence of the influences of editing the *SH4* and *qSH1* genes in weedy rice plants (Figure 6a–d). The intersecting section on the pedicels of the seed-shattering weedy rice control “C9” showed very smooth surfaces after the seeds were spontaneously shattered because of the fragile abscission layer with the hydrolyzed cell wall tissues (Figure 6a). In contrast, the pedicels of the *SH4*, *qSH1* (e.g., S5 and Q5), or double gene (e.g., SQ1)-edited plants showed rough intersecting section surfaces after seeds were removed from the plants by force due to the absence of abscission layers (Figure 6b–d). Overall, these results indicate a close association between editing the *SH4* and *qSH1* genes and the formation of abscission layers.

### 3.2. Transcriptomic Analyses of SH4- and qSH1-Edited Weedy Rice Plants and the Control

To determine the differential expression profiles of weedy rice genes across the entire genomes that were affected by the editing of seed-shattering *SH4* and *qSH1* genes, we performed comparative transcriptomic analyses between the seed-shattering weedy rice control “C9” and *SH4*, *qSH1*, and *SH4/qSH1* gene-edited plants at different stages of seed development. The clean reads were mapped to the reference genome, and expression levels of 32,552 genes were obtained. Of these, 1167, 948, and 6007 DEGs (differentially expressed genes) were identified in the *SH4*-gene edited plant at the preliminary, middle, and late stages of seed development, respectively. The *qSH1* gene-edited plant showed 2379 DEGs at the preliminary stage, 355 DEGs at the middle stage, and 6251 DEGs at the late stage of seed development. There were 3209, 409, and 4884 DEGs in the double *SH4*/*qSH1* gene-edited plant at the preliminary, middle, and late stages of seed development, respectively. These results suggest that *SH4*, *qSH1*, or *SH4/qSH1* gene editing affected gene expression extensively and particularly at the late stage. GO enrichment analyses showed that DEGs in *SH4* and *qSH1* gene-edited plants at the late stage were enriched, particularly corresponding to ABA; DEGs in the *SH4*/*qSH1* gene-edited plant at the late stage were enriched, also corresponding to the abscisic acid-activated signaling pathway and abscisic acid binding.

Interestingly, our results based on the transcriptomic analyses showed a large number of shattering-controlling genes (e.g., *SH4*, *OsSH1*, *OsSH15*, *qSH1*, *SH5*, and *SH8*), ABA biosynthesis-associated genes (e.g., *ZEP*, *NCED1.1*, *NCED1.2*, *SDR1*, and *SDR*), and ABA signaling transduction-related genes (e.g., *PYL3*, *PYL4*, *PP2C39*, *SAPK5*, *BZIP23*), and the cell-wall hydrolysis-associated genes (e.g., *PME34*, *PG*, *XTH28*, and *MAN4*) had significant repression at the late stage of seed development in the gene-edited weedy rice plants, such as S5, Q5, and SQ1, compared with the seed shattering weedy rice control “C9” (Figure 7). However, upregulation was detected for the *CPL1* gene, a quantitative trait locus (QTL) that negatively regulates seed shattering in rice (Figure 7). These results suggest considerable repression or suppression in the pathways of ABA biosynthesis signaling and hydrolysis processes in the *SH4*, *qSH1*, and *SH4*/*qSH1* gene-edited plants.

### 3.3. Expression Patterns of the Seed Shattering-Associated Genes and the Content of Plant Hormones

To confirm the expression of identified genes closely associated with the ABA biosynthesis and cell-wall hydrolysis processes, we further analyzed the expression of these genes using the RT-qPCR method. The results generally supported our transcriptomic data, in which the majority of the tested genes, such as *SH4*, *OsSH1*, *OsSH15*, *qSH1*, *SH5*, *SH8*, *PYL3*, *PYL4*, *PP2C39*, *SAPK5*, *BZIP23, PME34*, *PG*, *XTH28*, and *MAN4,* were associated with ABA hormone biosynthesis and cell-wall hydrolysis, showing significantly reduced expression at the late stage of seed development in the S5, Q5, and SQ1 plants (*p* < 0.05, Figure 8, Figure 9, Figure 10 and Figure 11). However, inconsistent results were detected for *CPL1* expression, showing a considerably decreased level of expression in all gene-edited weedy rice plants by RT-qPCR; nevertheless, the gene was upregulated in the transcriptomic analyses (see Figure 7a and Figure 8a). Altogether, these results confirmed that expression of these genes involved in seed-shattering control, ABA biosynthesis, ABA signaling, and cell-wall hydrolysis at the late stage of seed development was substantially repressed by editing of the *SH4*, *qSH1*, and *SH4*/*qSH1* genes.

To reveal the potential effect of editing the seed-shattering genes on the biosynthesis of ABA, seed development-related IAAs, and cytokinins, we focused on metabolome analysis for the relative content of plant hormones in the *SH4*-, *qSH1*-, and *SH4*/*qSH1*-edited weedy rice plants and the control “C9”. The results generated from metabolomic analyses indicated that S5 at the middle stage of the seed development and Q5 and SQ1 at the middle and later stages of the seed development had a significantly lower ABA content than the control plant “C9” (*p* < 0.05, Figure 12). However, no significant differences were detected in the other stages between gene-edited plants and the control plant “C9” (*p* ≥ 0.05, Figure 12). These results suggest that the decrease in ABA biosynthesis caused by the editing of these genes was one of the explanations for reduced seed shattering. The findings support the results of the gene expression analyses.

However, the editing of the *SH4* and *qSH1* genes did not show consistent and significant effects on the biosynthesis of IAAs (auxin a, auxin b) and cytokinins, although significant differences were detected in the relative contents of some of the IAAs and cytokinins (e.g., tZ, cZ, tZR, and cZR; Appendix A). These results suggest the complicated influences of IAAs and cytokinins by editing the *SH4* and *qSH1* genes.

## 4. Discussion

### 4.1. Editing the SH4 or qSH1 Gene Substantially Reduces Seed Shattering with Limited Influences on Seed Size-Related Traits of Weedy Rice

Our results from this study indicated that editing a single seed-shattering gene, either *SH4* or *qSH1*, substantially reduced the seed shattering of weedy rice plants that initially had strong seed shattering before gene editing. Similarly, editing the two *SH4* and *qSH1* genes at the same time also substantially reduced the seed shattering of weedy rice plants. Compared with the parental weedy rice line (“C9”), in which the SSI was deliberately determined as 100%, the *SH4* gene-edited weedy rice lines showed an average of 76.4% reduced ratios of seed shattering, and the *qSH1* gene-edited weedy rice lines showed an average of 87.8% reduced ratios of seed shattering. Likewise, the double gene-edited weedy rice lines showed an average of 74.8% reduced ratios of seed shattering. Noticeably, the average values of reduced seed shattering in both the single and double gene-edited weedy rice lines were much lower than that of cultivated rice; the seed persistence control showed only an average of 46.3% lower ratios of seed shattering. These results clearly answered our first question in the Introduction section regarding the consequences of editing the *SH4* and *qSH1* genes, in which significantly reduced seed shattering in weedy rice plants was evidently presented.

The microscopy comparisons based on the laser scanning confocal examination of rice pedicels at the longitudinal sections of the seed-shattering control line (“C9”) and the gene-edited lines clearly indicated the absence or weakness of the abscission layer tissues. This result provided direct evidence to demonstrate that editing the *SH4* or *qSH1* gene can considerably alter the abscission layers, which is known to be closely associated with the biosynthesis of ABA, as reported by many previous studies concerning seed shattering in rice [36,37,38,39]. Altogether, the results confirmed that editing a seed-shattering gene, either *SH4* or *qSH1*, and probably also other seed-shattering genes reported in rice, will significantly reduce seed shattering in weedy rice [36,40]. Undoubtedly, it is possible to reduce seed shattering of wild/weedy rice plants through genetic engineering, such as editing the seed-shattering genes. Therefore, this technology can be applied to develop a transgene mitigation system with a tightly linked tandem “safe-box” in rice genetic improvement and wide application in GE rice varieties.

In previous studies, the *SH4* and *qSH1* genes were mapped on chromosomes 4 and 1, respectively [36,41,42]. These genes are considered the two major genes regulating seed shattering in species of the genus *Oryza* (Poaceae), including cultivated rice and wild/weedy *Oryza* relatives. The point mutations, including insertion or deletion of two to three nucleotides in the two genes, resulted in a deficiency in the abscission layers, reducing the seed-shattering ability in rice and its wild/weedy relatives [36,41,42,43]. In a previous study, we reported the transfer of silenced miRNAs targeting the altered expression of the *SH4* gene into a cultivated rice variety and produced a GE rice line with reduced expression of the *SH4* gene. To mimic the transgene flow from GE rice to weedy rice, we produced hybrid progeny using this GE rice line (as a pollen donor) crossed with a few weedy rice populations (as pollen recipients). As expected, the obtained hybrid progeny (F_1_ and F_2_) with reduced expression of the *SH4* gene demonstrated significantly reduced seed shattering [25]. That study successfully demonstrated the possibility of developing a transgene “safe-box” by manipulating the seed-shattering genes in rice. Compared with previous studies, this study indicated a much stronger power of using gene editing for the reduction of seed shattering in rice. Altogether, this finding confirms the potential applications of gene editing biotechnology in developing a new generation of “safe-boxes” tightly linked with target transgenes to mitigate the adverse impact of escaped transgenes in GE rice varieties through genetic engineering of the seed-shattering genes.

In addition, our results showed that the editing of a single gene, either the *SH4* or *qSH1* gene controlling seed shattering, did not have considerable and consistent negative fitness effects on some of the seed size-related traits, particularly 1000-seed weight and seed width, although the length of seeds showed a significant reduction in some of the gene-edited weedy rice lines. These results suggest that the editing of the seed-shattering genes may not significantly affect grain yield-associated traits in rice, at least in some of the gene-edited lines, given that these lines showed considerable variation in the seed size-related traits. Therefore, it is possible to select lines with significantly reduced seed shattering but no significant influences on their grain yield after editing the seed-shattering genes. Such a mitigating system of gene-edited rice plants can serve as an ideal “safe-box” for transgenic rice varieties that have normal grain yield for rice production but have maintained or enhanced seed persistence. If the transgene(s) together with the “safe-box” transfers to weedy rice plants through gene flow, the transgenes will not be able to quickly spread in weedy rice populations simply because of enhanced seed persistence [25,27]. Therefore, the results obtained in this study demonstrate the great potential for developing a mitigating system or technology by editing seed-shattering genes, such as *SH4* or *qSH1*, which can be applied as a “safe-box” in the commercial cultivation of GE rice to reduce the adverse environmental impacts caused by transgene flow to wild/weedy relatives of GE rice.

However, further studies regarding the influences of editing the seed-shattering genes on other agronomic traits, particularly the key traits closely associated with rice grain production, are urgently needed to ensure the benefits obtained by genetic engineering. Only when the reduced seed shattering or increased seed persistence does not affect the yield-related traits of rice plants does the breeding of rice varieties containing the edited seed-shattering gene, as the “safe-box,” become valuable and applicable. In addition, we would not recommend editing more than one seed-shattering gene for developing the “safe-box” as the transgene mitigation strategy because double gene editing (*SH4*/*qSH1*) did not have better effects on the reduction of seed shattering than single gene editing (*SH4* or *qSH1*). Furthermore, double gene editing may potentially have somehow greater negative effects (fitness cost) on rice plants based on our results, in which we detected some degree of reduction in the seed size-related traits (e.g., 1000-seed weight) in all the double gene-edited lines.

### 4.2. Editing SH4 and qSH1 Genes Regulates the Expression of Seed Shattering-Related Genes and Biosynthesis of the Key Hormone in Weedy Rice

Our results based on the transcriptomic analyses and RT-qPCR expression showed significant downregulation of most examined genes closely associated with seed shattering (e.g., *SH4*, *OsSH1*, *OsSH15*, *qSH1*, *SH5*, and *SH8*), ABA biosynthesis (e.g., *ZEP*, *NCED1.1*, *NCED1.2*, *SDR1*, and *SDR*), ABA signaling transduction (e.g., *PYL3*, *PYL4*, *PP2C39*, *SAPK5*, and *BZIP23*), and cell wall hydrolysis (e.g., *PME34*, *PG*, *XTH28*, and *MAN4*) in *SH4* and *qSH1* gene-edited weedy rice lines. Consistent changes of reduced expression of the above-examined genes evidently indicated that the observed phenomenon of reduced seed shattering in the *SH4* and *qSH1* gene-edited weedy rice lines is most likely attributable to considerable suppression of ABA biosynthesis and signaling. In addition, the reason for reduced seed shattering is also due to reduced bioactivities in the cell wall hydrolysis pathway resulting from gene editing. In other words, editing of seed-shattering genes, including *SH4* and *qSH1*, can considerably reduce the biosynthesis and signaling of the ABA hormone and the bioactivities in the cell wall hydrolysis pathway, resulting in significantly reduced seed shattering in weedy rice. This prediction or observation can also be strongly supported by the microscopy results of the weakness and absence of the abscission layer tissues in the pedicels of gene-edited weedy rice lines.

Previous studies on genetic mapping of the seed-shattering regulatory network in rice and its wild relatives involved multiple positive-regulating QTLs, such as *SH4*, *SHAT1*, *OsSH1*, *OsSH15*, *qSH1*, *SH5*, and *SH8* [44,45,46,47]. This network regulating the plant hormone signaling pathways includes ABA activation and auxin (IAA) inhibition in abscission layer tissues, which triggers the process of cell wall hydrolysis and then causes the breakage of the pedicels connected to seeds [37,48,49]. Obviously, these performances are regulated by a group of hydrolases, such as polygalacturonase and endo-b-glucanase. Therefore, we proposed that the negative responses or downregulations of the above QTLs, including reduced ABA biosynthesis, ABA signaling, and cell wall hydrolysis expression, together caused reduced seed shattering in the *SH4*- and *qSH1*-edited weedy rice lines. However, more detailed knowledge of the transcriptomic variation in the seed-shattering regulation gene network triggered by editing of the key seed-shattering genes would provide new insights into the regulatory mechanisms associated with seed-shattering characteristics in weedy rice and other *Oryza* species.

In addition, our analysis of phytohormones supported the variation pattern of gene expression in the ABA pathway, in which the ABA content showed considerable decreases in the *SH4* and *qSH1* gene-edited weedy rice lines, as well as in the double gene-edited lines compared with their parental weedy rice control (“C9”). However, no consistent variation was detected in the contents of other key plant hormones, such as IAA and cytokinins, in all the *SH4* and *qSH1* gene-edited weedy rice lines in our study. Noticeably, previous studies considered that the changes of some key plant hormones, mostly the IAAs and cytokinins, would have great influences on the shapes and weights in rice seeds, based on the results of increased IAA content and decreased cytokinins during endosperm development in plant seeds, which might cause a reduction in seed size [50,51,52]. However, such a pattern of hormone and seed size changes was not observed in this study regarding the single *SH4* or *qSH1* gene-edited weedy rice lines showing substantially reduced seed shattering. Our results suggest that biosynthesis of the examined plant hormones, such as IAAs and cytokinins, may not have a significant influence on the editing of the *SH4* and *qSH1* genes in weedy rice. Therefore, we consider that editing of the *SH4* or *qSH1* gene to reduce seed shattering, as a “safe-box” or transgene mitigator, may not have considerably influenced the seed size-related traits in the resulting weedy rice or rice lines. Whether or not this consideration is solid still needs more detailed investigation for verification. At least, it is possible to select the gene-edited lines that do not have reduced or even increased seed production, given the considerable variation in the seed size-related traits among the lines.

### 4.3. Implications for Environmental Biosafety and Weedy Rice Control through Mitigating Genes Transmitted from Cultivated Rice

Rice is one of the world’s most important crops, providing staple food for more than one-half of the world’s population [53]. Therefore, rice serves as an important food supply, ensuring world food security. With the rapid increase in the global human population, the demand for rice production is also increasing, which requires advanced biotechnologies for rice genetic improvement to increase rice grain production. However, potential environmental biosafety concerns hinder the commercial cultivation of GE rice varieties. The possible adverse environmental impacts caused by transgene flow from GE rice to its wild and weedy relative species are among these concerns [7,20,54]. Given that weedy rice co-occurs with its cultivated counterpart, it is nearly impossible to stop transgene flow from GE rice varieties to weedy rice populations [16,17]. Therefore, mitigating the function of the outflowing transgenes that convey selective advantages to weedy rice (even wild rice) will provide solutions for reducing or mitigating the potential environmental impact. This is because reduced seed shattering would not have any adverse impact on cultivated rice and even have increased value for cultivated rice to avoid grain yield losses during harvesting, but obviously have disadvantages for self-reproduction of weedy rice.

Our results from this study evidently demonstrated that editing the seed-shattering genes, including *SH4* and *qSH1*, can substantially reduce seed shattering or, in other words, increase the seed/grain persistence of weedy rice that originally had a very strong seed-shattering ability. The ratio of reduced seed shattering by gene editing was much greater than that in our previous study using the silenced *SH4* gene to reduce seed shattering [25]. The seed-shattering ability in the gene-edited weedy rice lines, particularly the *qSH1*-edited lines, was much lower than that in the cultivated rice control (‘Shouxiang-1′). Therefore, it is possible to establish a shattering gene editing-based transgene “safe-box” in the GE rice system to mitigate the potential adverse environmental impact caused by transgene outflow from GE rice to weedy or wild relative species in the genus *Oryza*. This transgene mitigator or “safe-box” system will have great value for the development and applications of GE rice having agronomic traits with strong natural selective advantages, such as herbicide resistance [7]. This gene editing-based transgene “safe-box” system can also be transferred to other GE crops having the same problems of transgene escape to their weedy and wild relatives by reduced seed shattering or fruit fall in agroecosystems or natural ecosystems. This biosafety strategy can largely reduce or even avoid the adverse environmental impacts caused by transgene flow from GE crops, given that it is impossible to stop transgene flow, which is a part of the evolutionary process.

In addition, the results obtained from this study, in which a weedy rice line (“C9”) originally with strong seed shattering developed a substantial reduction in its seed-shattering ability after gene editing, can encourage the design of powerful techniques for weedy rice control, as we previously proposed [25]. Usually, seed shattering at maturity is the main reason for the infestation and outbreaks of weedy rice in rice fields. The more shattered seeds stored in the soil seed banks, the greater the scale of weedy rice populations that will be established in the next rice cultivation seasons, causing severe weed problems. Therefore, a strategic design to stop or substantially reduce seed shattering and to greatly reduce the quantity of weedy rice seeds in the soil seed banks can effectively control the infestation of weedy rice. For example, if a GE rice variety is developed by the gene-editing technique and carries the mitigator encoding traits with substantially reduced seed shattering, the gene flow from this GE rice variety to weedy rice populations co-occurring in the same rice fields will automatically occur in different generations.

Such recurrent crop-to-weed gene flow will incessantly “pump” the mitigator to weedy rice populations and substantially reduce seed shattering in weedy rice populations from one generation to another. Consequently, more non-shattering or persistent weedy rice seeds will be harvested from the rice plants, resulting in a decline in weedy rice populations in the rice fields. In the long run, the recurrent and spontaneous transmission of the edited gene(s) or mitigator encoding reduced seed shattering from cultivated rice varieties to weed rice populations will become very effective for weedy rice control, turning these troubles into benefits. The mitigator technique can either be adopted alone for weedy rice control, as mentioned above or can also be applied together with a transgene(s) as a “safe-box” for mitigating potential environmental impacts. For example, the transformation of the mitigator into a GE rice variety that is tightly linked with target transgenes such as herbicide resistance transgenes will serve as a “safe-box” to balance the benefit of the tightly linked target transgenes and, at the same time, mitigate the ability of competition and spread of crop–weed hybrid descendants for environmental biosafety purposes [26].

Whether or not the mitigator strategy can be successfully applied in GE rice production for the purpose of environmental biosafety or weed control depends on further investigation, evaluation, and optimization of the techniques. Increased knowledge, such as the extent of the crop–weed introgression, the co-evolution of seed shattering in cultivated and weedy rice, the change in fitness-related traits in gene-edited plants, and the gene-regulating network of seed shattering in rice, will open up a new dimension for us to mitigate the adverse environmental impacts of transgene flow and to effectively control weedy rice for GE rice production. Hopefully, this mitigating strategy in rice can also be expanded to other crop species.

## 5. Conclusions

In conclusion, we found that editing a single seed-shattering gene, either *SH4* or *qSH1*, and editing both of these genes substantially reduced the seed-shattering ability of the weedy rice control (“C9”) that originally had strong seed shattering. In addition, the single gene-edited weedy rice lines with reduced seed shattering did not show a consistent reduction in their seed size-related traits, particularly the 1000-seed weight (e.g., *qSH1* gene-edited lines), although the double gene-edited weedy rice lines showed some degrees of reduction. The reduced seed-shattering ability in the gene-edited weedy rice lines was closely linked with the weakness and absence of abscission layers, of which the decreased plant hormone ABA was detected. Further transcriptomic and RT-qPCR expression analyses demonstrated significant downregulation of most examined genes that were closely associated with ABA biosynthesis, ABA signaling transduction, and cell wall hydrolysis in the *SH4*-, *qSH1*-, and *SH4*/*qSH1*-edited weedy rice lines compared with the weedy rice control. These findings are very important not only for our deep insights into the underlying mechanisms of reduced seed shattering in weedy rice plants but also for encouraging practical applications in rice production. As a tightly linked tandem “safe-box” of a transgene with substantially reduced seed shattering, the gene editing-based system can help us to design strategies for mitigating the potential adverse environmental impacts caused by transgene flow from GE rice varieties to weedy and wild rice relatives. In addition, this gene editing-based system can be applied to manage the infestation of weedy rice by acquiring the mitigator from GE rice cultivars, such as continued “pumping” seed-shattering reduction genes into weedy rice from GE rice varieties through natural gene flow.

## Figures and Tables

**Figure 1 biology-11-01823-f001:**
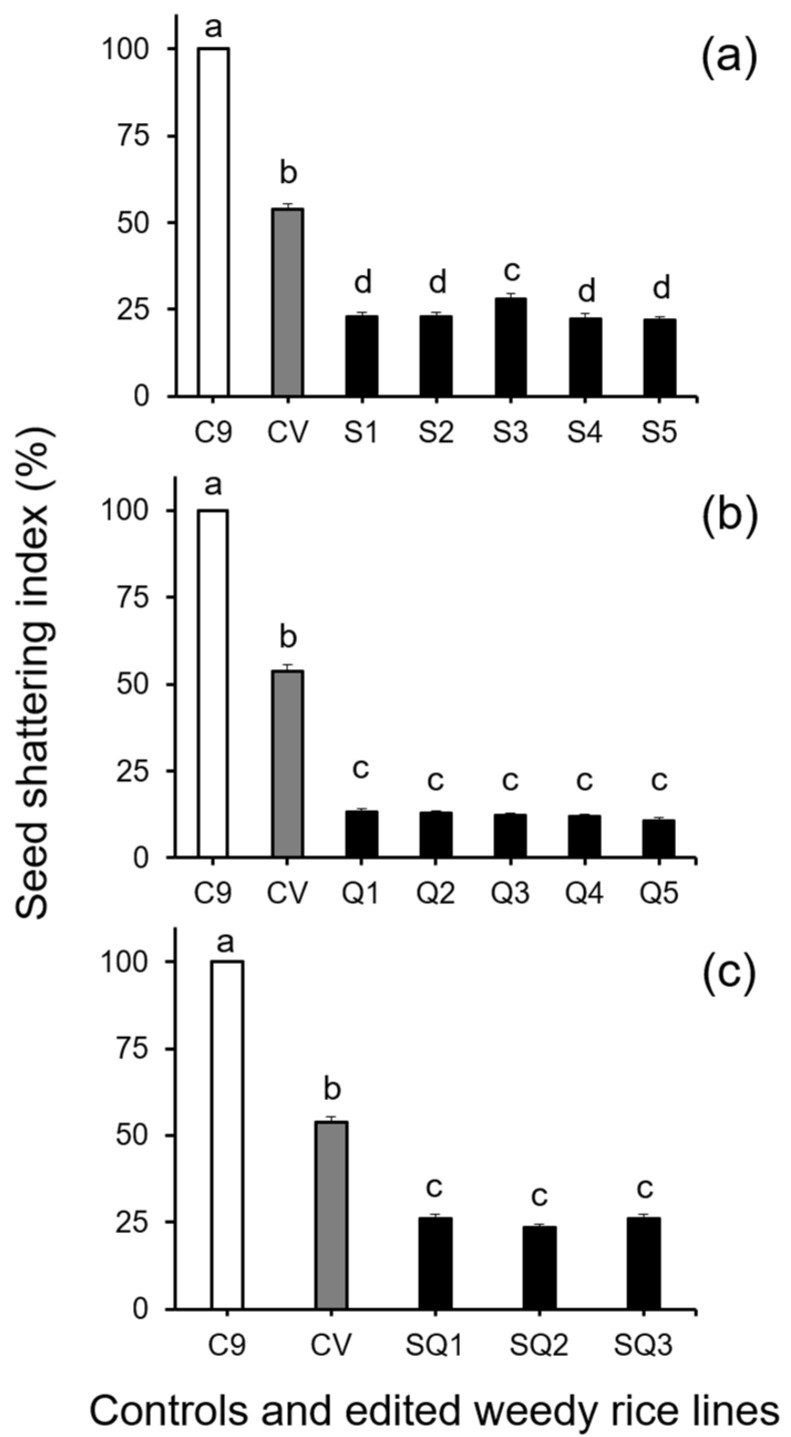
Average seed shattering index (SSI, n = 100 replicates) of the weedy rice parent line (“C9”, white columns), cultivated rice line (CV, gray columns), and the gene-edited weedy rice lines (black columns). (**a**), The weedy rice and cultivated rice controls, and *SH4* gene-edited lines (S1–S5); (**b**), weedy rice and cultivated rice controls, and *qSH1* gene-edited lines (Q1–Q5); (**c**), weedy rice and cultivated rice controls, and *SH4*/*qSH1* gene-edited lines (SQ1–SQ3). Bars represent the standard errors. Different small letters indicate significant differences at *p*-value < 0.05 based on Duncan’s multiple range test [33].

**Figure 2 biology-11-01823-f002:**
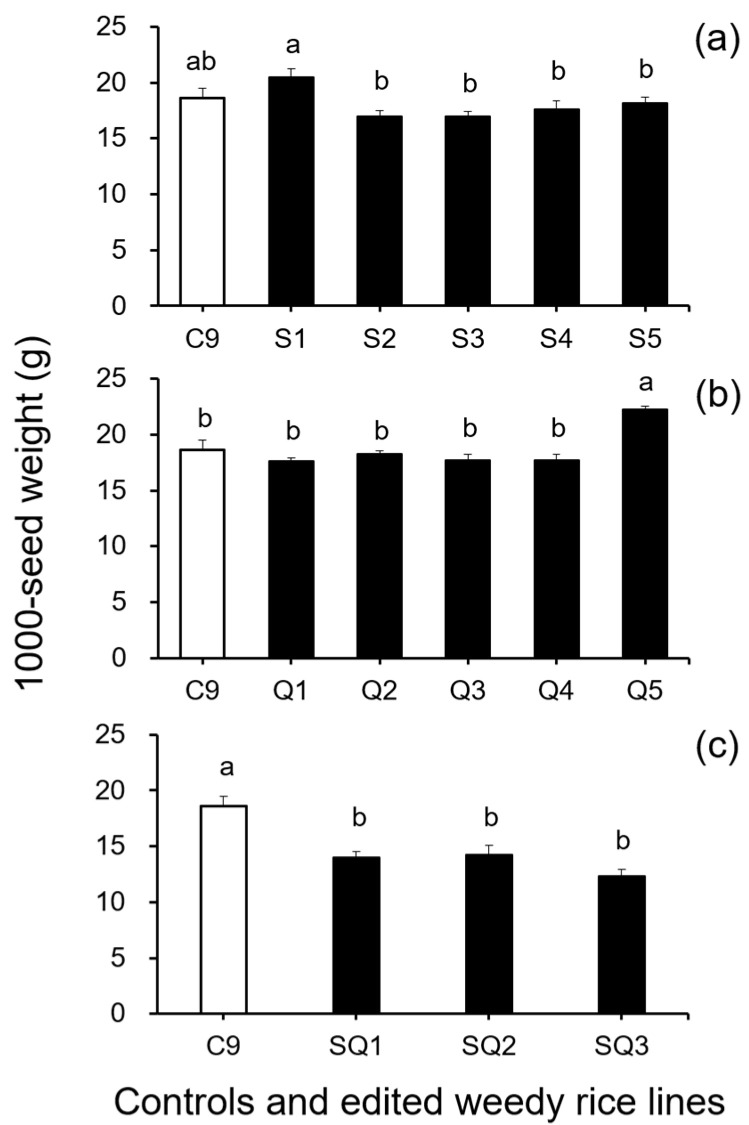
Average 1000-seed weight (n = 5 replicates) of the weedy rice parent line (“C9”, white columns), cultivated rice line (CV, gray columns), and the gene-edited weedy rice lines (black columns). (**a**), The weedy rice control and its *SH4* gene-edited lines (S1–S5); (**b**), the weedy rice control and its *qSH1* gene-edited lines; (**c**), the weedy rice control and its *SH4/qSH1*-gene edited lines. Bars represent the standard errors. Different small letters indicate significant differences at *p*-value < 0.05 based on Duncan’s multiple range test [33].

**Figure 3 biology-11-01823-f003:**
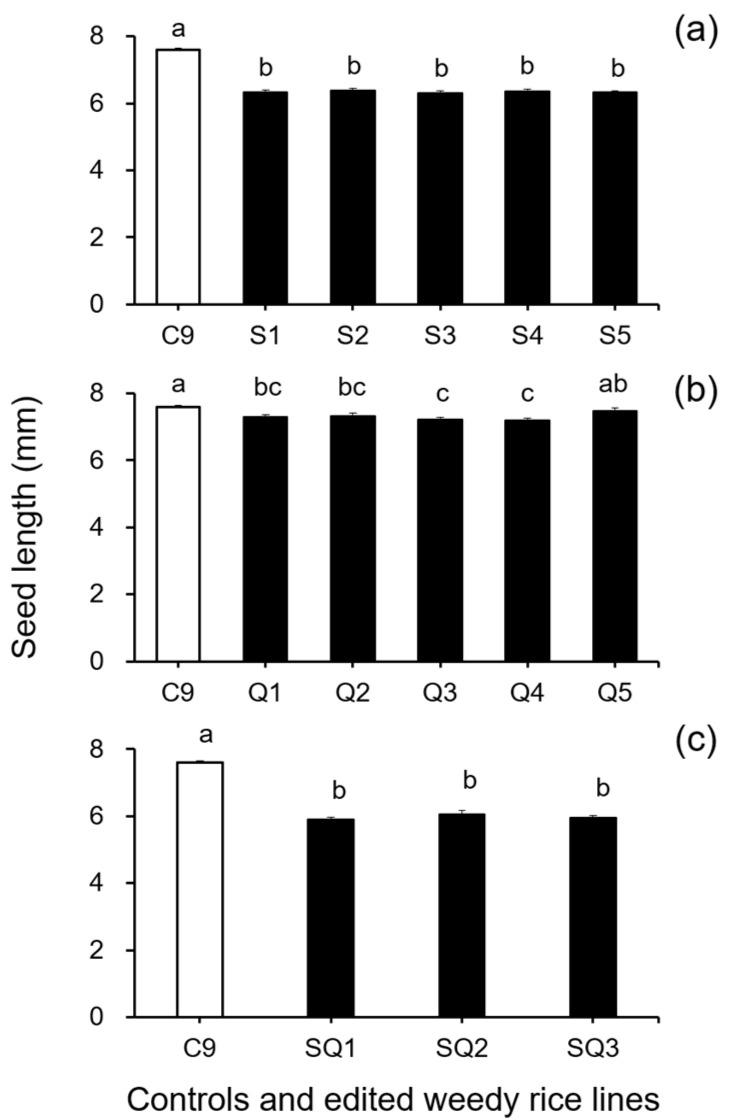
Average seed length (n = 20 replicates) of the weedy rice parent line (“C9”, white columns) and the gene-edited weedy rice lines (black columns). (**a**), The weedy rice control and its *SH4* gene-edited lines (S1–S5); (**b**), the weedy rice control and its *qSH1* gene-edited lines (Q1–Q5); (**c**), the weedy rice control and its *SH4/qSH1* gene-edited lines (SQ1–SQ3). Bars represent standard errors. Different small letters indicate significant differences at *p*-value < 0.05 based on Duncan’s multiple range test [33].

**Figure 4 biology-11-01823-f004:**
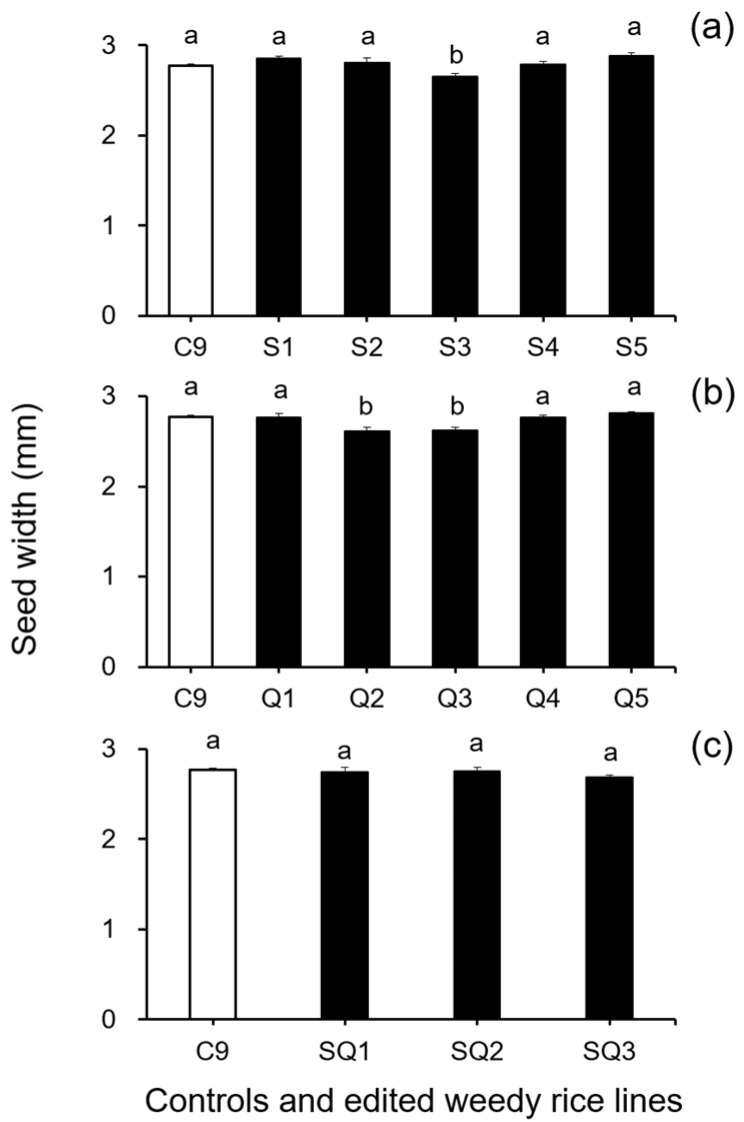
Average seed width (n = 20 replicates) of the weedy rice parent line (“C9”, white columns) and the gene-edited weedy rice lines (black columns). (**a**), The weedy rice control and its and *SH4*-edited lines (S1–S5); (**b**), the weedy rice control and its *qSH1*-edited lines (Q1–Q5); (**c**), the weedy rice control and its *SH4*/*qSH1*-edited lines (SQ1–SQ3). Bars represent standard errors. Bars represent standard errors. Different small letters indicate significant differences at *p*-value < 0.05 based on Duncan’s multiple range test [33].

**Figure 5 biology-11-01823-f005:**
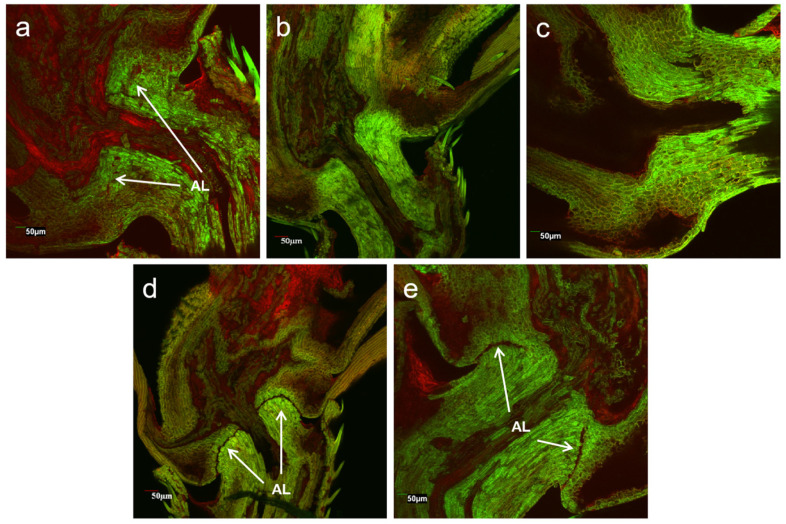
Laser scanning confocal microscopy showing the longitudinal sections of pedicels of the *SH4* gene-edited line S5 (**a**); the *qSH1* gene-edited line Q5 (**b**); the double gene (*SH4*/*qSH1*)-edited line SQ1 (**c**); the weedy rice parental line “C9” (**d**); and the cultivated rice control (**e**). AL indicates the abscission layers.

**Figure 6 biology-11-01823-f006:**
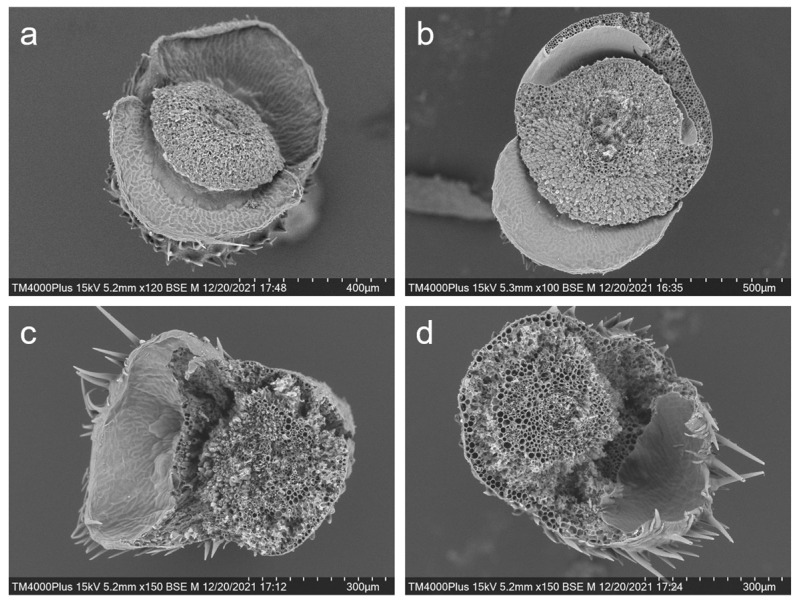
Scanning electron microscopy showing the intersecting surface of the pedicels of the weedy rice prenatal line “C9” (**a**); the *SH4* gene-edited line S5 (**b**); the *qSH1* gene-edited line Q5 (**c**); and the double gene (*SH4*/*qSH1*)-edited line SQ1 (**d**).

**Figure 7 biology-11-01823-f007:**
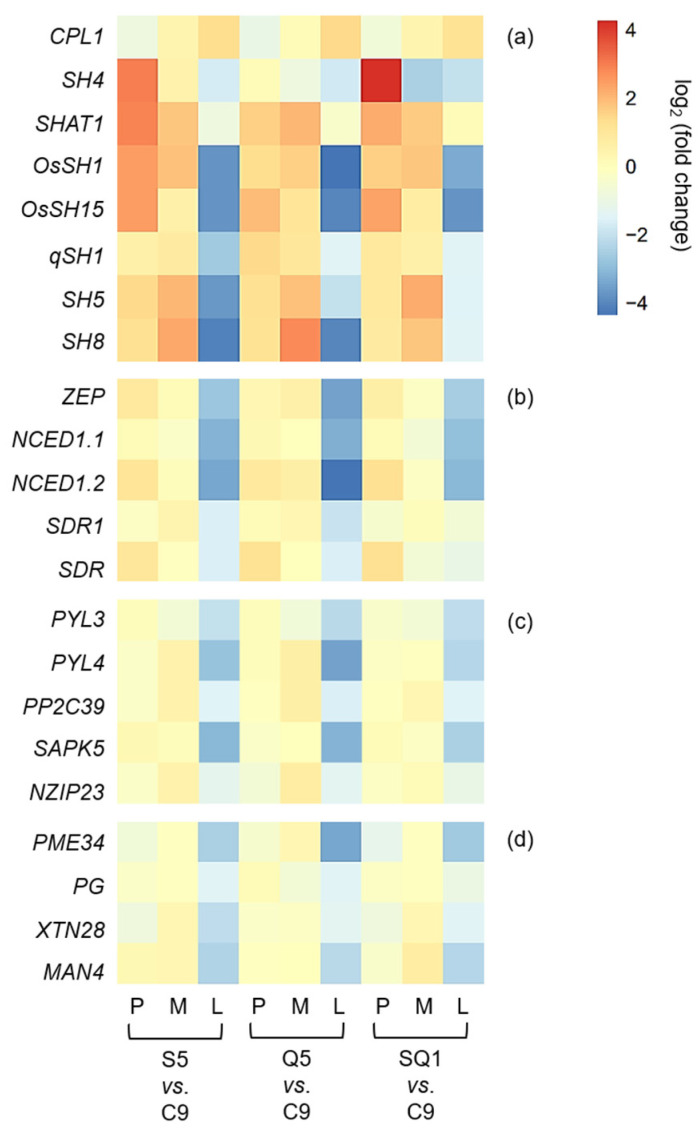
Heatmaps showing the transcription levels of different genes in the gene-edited weedy rice lines S5, Q5, and SQ1. (**a**) Genes associated with seed shattering; (**b**) genes encoding the key enzyme of abscisic acid (ABA) biosynthesis; (**c**) genes related to the ABA-signaling pathway; and (**d**) genes encoding cell-wall hydrolases. Comparisons of the gene transcription levels were made between the weedy rice parental line “C9” and its gene-edited lines. The letters P, M, and L indicate the preliminary, middle, and later stages of seed development, respectively.

**Figure 8 biology-11-01823-f008:**
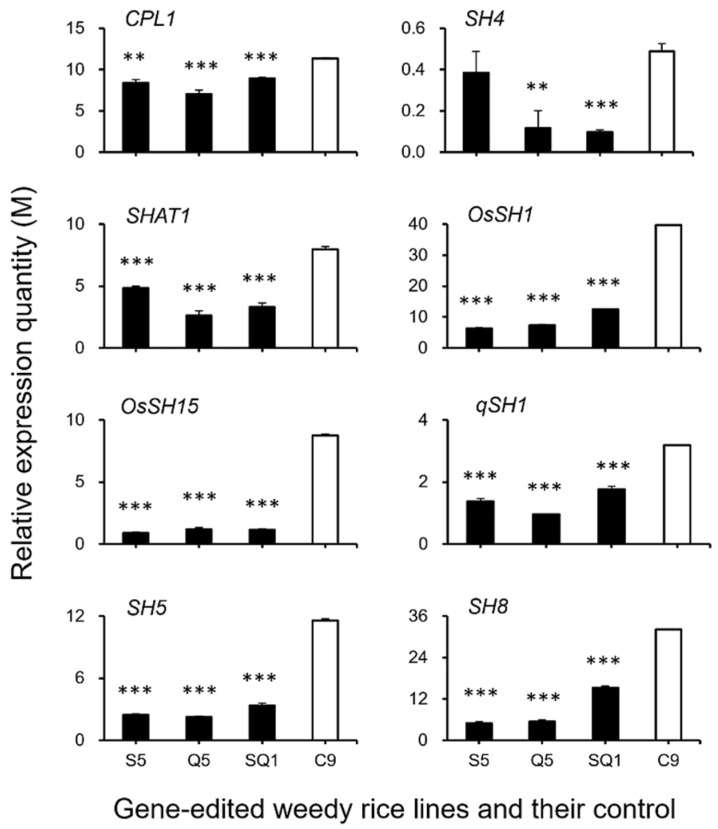
Relative expression quantities (n = 3 replicates) of the genes controlling seed shattering (*CPL1*, *SH4*, *SHAT1*, *OsSH1*, *OsSH15*, *qSH1*, *SH5*, and *SH8*). The comparisons were made between the gene-edited lines S5, Q5, SQ1, and the weedy rice parental line “C9” based on the independent-sample Student’s *t*-test. Bars represent standard errors. ** *p* < 0.01, *** *p* < 0.001. M indicates multiples of the relative expression quantity (see Materials and Methods).

**Figure 9 biology-11-01823-f009:**
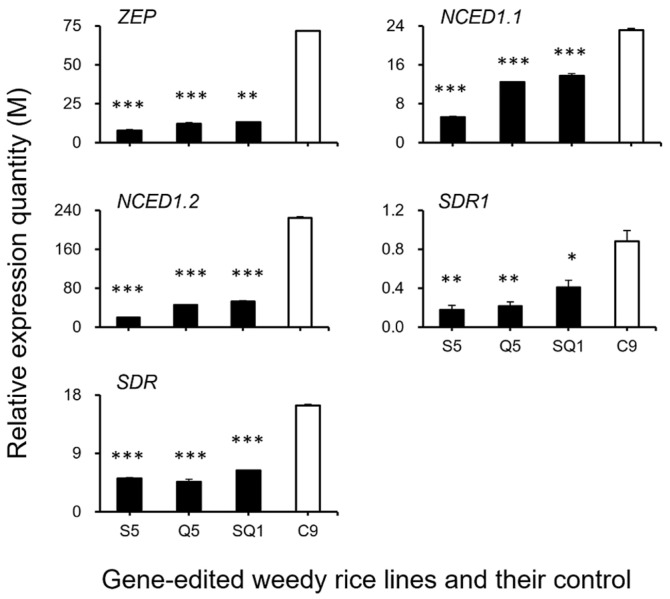
Relative expression quantities (n = 3 replicates) of the genes (*ZEP*, *NCED1.1*, *NCED1.2*, *SDR1*, and *SDR*) encoding the key enzymes for ABA (abscisic acid) biosynthesis. Comparisons were made between the gene-edited lines, S5, Q5, and SQ1, and the weedy rice control “C9” based on the independent-sample Student’s *t*-test. Bars represent standard errors. * *p* < 0.05, ** *p* < 0.01, *** *p* < 0.001. M indicates multiples of the relative expression quantity (see Materials and Methods).

**Figure 10 biology-11-01823-f010:**
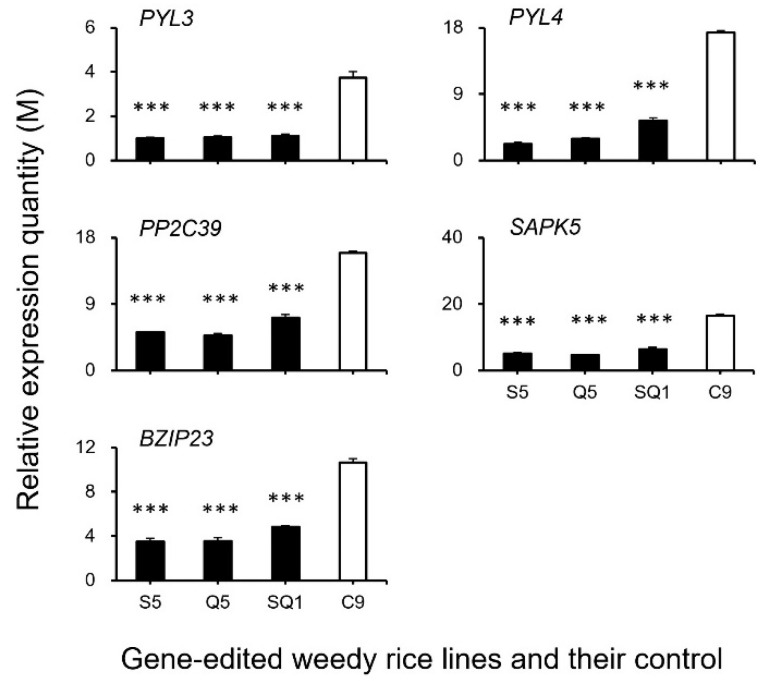
Relative expression quantities (n = 3 replicates) of the genes associated with the ABA (abscisic acid) signaling pathway (*PYL3*, *PYL4*, *PP2C39*, *ASPK5*, and *BZIP23*). Comparisons were made between the gene-edited lines, S5, Q5, and SQ1, and the weedy rice control “C9” based on the independent-sample Student’s *t*-test. Bars represent standard errors. *** *p* < 0.001. M indicates multiples of the relative expression quantity (see Materials and Methods).

**Figure 11 biology-11-01823-f011:**
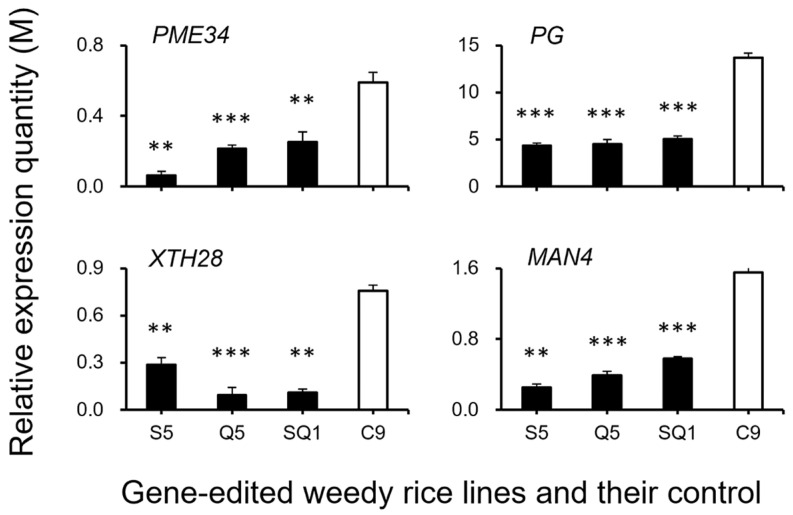
Relative expression quantities (n = 3 replicates) of the genes (*PME34*, *PG*, *XTH28*, and *MAN4*) encoding the hydrolases. Comparisons were made between the gene-edited lines, S5, Q5, and SQ1, and their weedy rice control “C9” based on the independent-sample Student’s *t*-test. Bars represent standard errors. ** *p* < 0.01, *** *p* < 0.001. M indicates multiples of the relative expression quantity (see Materials and Methods).

**Figure 12 biology-11-01823-f012:**
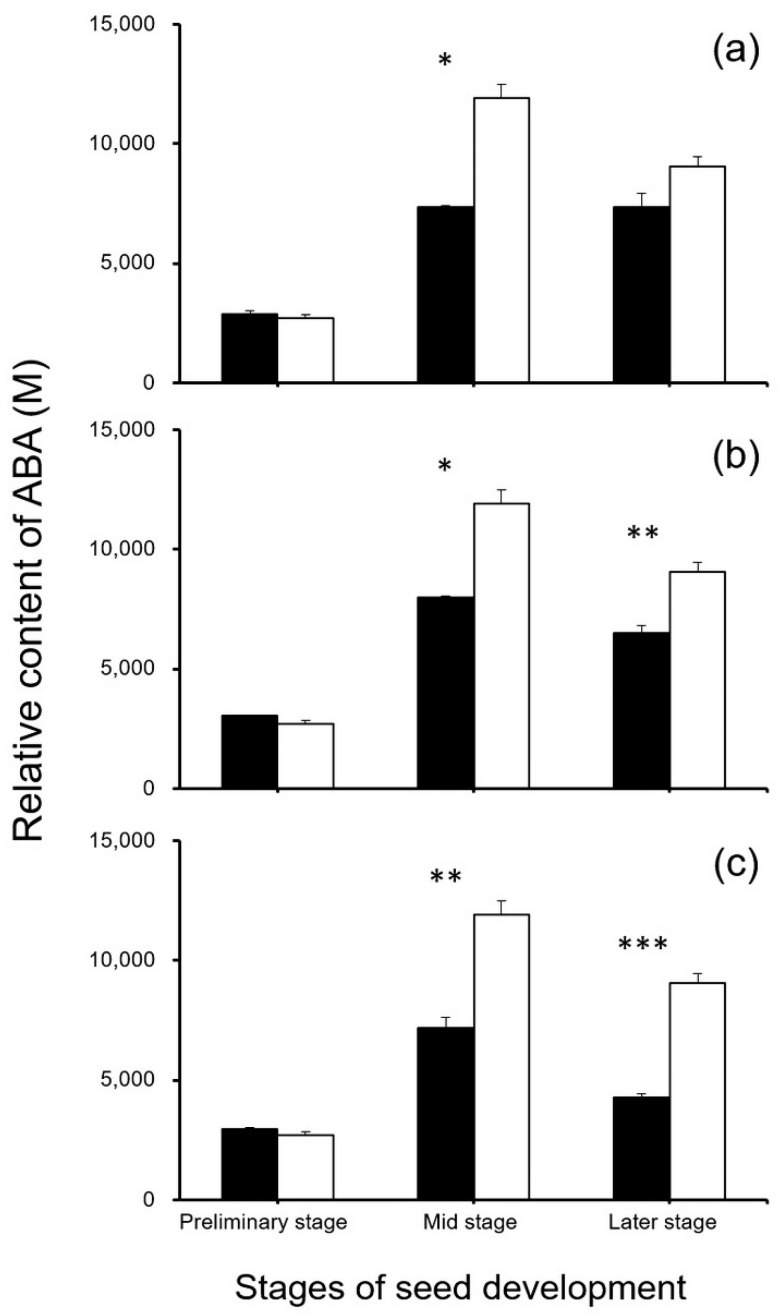
Comparison of relative ABA (abscisic acid) content (n = 3 replicates) in the gene-edited lines S5, Q5, and SQ1 (black columns) and the weedy rice parental line “C9” (white columns) at the preliminary, middle, and later stages of seed development, based on the independent-sample Student’s *t*-test. Bars represent standard errors. * *p* < 0.05, ** *p* < 0.01, *** *p* < 0.001. M indicates multiples of the relative expression quantity (see Materials and Methods).

**Table 1 biology-11-01823-t001:** Identity (ID) of the *SH4*-, *qSH1*-, and *SH4*/*qSH1*-edited weedy rice lines and their variations in the edited locus during gene editing.

Edited Gene	ID of Gene-Edited Lines	Variation on the Edited Locus	
		**SH4-TL-a**	**SH4-TL-b**
	S1	7 bp deleted	15 bp deleted
	S2	245 bp deleted	No variation
*SH4*	S3	T added	No variation
	S4	A added	No variation
	S5	G deleted	No variation
		**qSH1-TL-a**	**qSH1-TL-b**
	Q1	564 bp deleted	No variation
	Q2	T added	A added
*qSH1*	Q3	T added	C added
	Q4	28 bp deleted	C deleted
	Q5	No variation	C deleted
		**SH4-TL-a**	**qSH1-TL-b**
	SQ1	439 bp deleted	C deleted
*SH4*/*qSH1*	SQ2	153 bp deleted	A added
	SQ3	10 bp deleted	T added

**Table 2 biology-11-01823-t002:** One-way ANOVA (analysis of variance) for the effect of *SH4*, *qSH1*, double gene (*SH4/qSH1*)-edited, and different gene-edited lines (plants) on seed shattering- and seed size-related traits. A comparison was made with the weedy rice line “C9” (control).

Factor	Df ^1^	Traits			
		Seed Shattering Index (%)	1000-Seed Weight (g)	Seed Length (mm)	Seed Width (mm)
		*F*-Value	*F*-Value	*F*-Value	*F*-Value
Editing the *SH4* gene	1	3017.3 ***	0.7	410.6 ***	0.3
*SH4*-gene edited lines	4	3.3 *	4.7 **	0.1	4.2 **
Editing the *qSH1*-gene	1	12,595.0 ***	0.0	19.8 ***	3.4
*qSH1*-gene edited lines	4	1.5	21.9 ***	1.5	5.8 ***
Double editing (*SH4/qSH1*)	1	3179.1 ***	46.4 ***	438.7 ***	1.5
*SH4/qSH1*-gene edited lines	3	1.2	2.2	0.823	0.6

^1^ Df, degrees of freedom. * *p* < 0.05; ** *p* < 0.01; *** *p* < 0.001.

## Data Availability

All the data were presented in the article.

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
