# Peer review of "Reducing Seed Shattering in Weedy Rice by Editing SH4 and qSH1 Genes: Implications in Environmental Biosafety and Weed Control through Transgene Mitigation"

_biology, 2022, doi:10.3390/biology11121823_

Round 1
Reviewer 1 Report
Mitigating the possible adverse environmental impacts caused by transgene flow from genetically engineered (GE) crops to their wild/weedy relatives is an ideal strategy to resolve biosafety problems. The authors edited the seed-shattering genes of a weedy rice. Transgenic plant exhibited substantially reduced seed shattering and seed-size related traits. In addition, the authors also found that reduced seed shattering was closely linked with the vanishment of abscission layers and reduced abscission acid. The present is interesting very much but some data should be improved.
In Figure 12, the units of the vertical coordinates need to be written out, and the “*”should be marked on the corresponding column.
Author Response
POINT-BY-POINT RESPONSES TO REVIEWER #1
General comments to the Authors:
Mitigating the possible adverse environmental impacts caused by transgene flow from genetically engineered (GE) crops to their wild/weedy relatives is an ideal strategy to resolve biosafety problems. The authors edited the seed-shattering genes of a weedy rice. Transgenic plant exhibited substantially reduced seed shattering and seed-size related traits. In addition, the authors also found that reduced seed shattering was closely linked with the vanishment of abscission layers and reduced abscission acid. The present is interesting very much but some data should be improved.
Response – We indeed appreciate the positive comments from the reviewer, we have largely improved our manuscript accordingly.
Comment-1. In Figure 12, the units of the vertical coordinates need to be written out, and the “*” should be marked on the corresponding column.
Response – Thanks a lot for the reminding, we have added the units (M, which is determined as the multiple quantity (M) relative to the SH4 gene expression in the weedy rice control “C9”) of the vertical coordinates in Figure 12 and other relevant Figures. We also included relevant explanation of the unit in the Material and Methods section. In addition, we have marked the “*” on the corresponding columns.

Reviewer 2 Report
In the study, two genes are edited in weedy rice to reduce seed loss, which is meaningful to practical applications for mitigating environmental impact caused by transgene flow and for controlling infection of weedy rice. I have no objection to the article's explanation of the physiological characteristics of reduced seed abscission through microstructure observation, transcriptome and hormone determination.
But I want to know the probability of these two genes being transferred from rice to weedy rice. If the probability is not high, it will lose its practical production significance.
Author Response
POINT-BY-POINT RESPONSES TO REVIEWER #2
General comments to the Authors:
In the study, two genes are edited in weedy rice to reduce seed loss, which is meaningful to practical applications for mitigating environmental impact caused by transgene flow and for controlling infection of weedy rice. I have no objection to the article's explanation of the physiological characteristics of reduced seed abscission through microstructure observation, transcriptome and hormone determination.
Response – We indeed appreciate the positive comments from the reviewer about our work.
Comment-1. But I want to know the probability of these two genes being transferred from rice to weedy rice. If the probability is not high, it will lose its practical production significance.
Response – We understand this concern. Actually, cultivated rice and weedy rice are conspecific, meaning that they belong to the same species and can have sexual crosses freely in nature. In our previous work, we have successfully made the transgenic rice containing the silenced SH4 gene that was transferred to weedy rice by sexual crosses and substantially reduced seed shattering of F1 and F2 hybrid progeny with the transgenic rice lines. Please see the reference below:
Huanxin Yan, Lei Li, Ping Liu et al. 2017. Reduced weed seed shattering by silencing a cultivated rice gene: strategic mitigation for escaped transgenes. Transgenic Research, 26: 465–475.
In this work, we directly produce seed-shattering reduced weedy rice by editing the SH4 and qSH1 genes in weedy rice, to prove that editing the seed-shattering genes is working.
In addition, following your suggestion, we have sent our manuscript to a English Editing Company to revise the English of our manuscript.

Round 2
Reviewer 2 Report
The author has made appropriate revisions in this article.